# Data Generation Scheme for Thermal Modality with Edge-Guided Adversarial Conditional Diffusion Model

## ABSTRACT

In challenging low-light and adverse weather conditions, thermal vision algorithms, especially object detection, have exhibited remarkable potential, contrasting with the frequent struggles encountered by visible vision algorithms. Nevertheless, the efficacy of thermal vision algorithms driven by deep learning models remains constrained by the paucity of available training data samples. To this end, this paper introduces a novel approach termed the edge-guided conditional diffusion model (ECDM). This framework aims to produce meticulously aligned pseudo thermal images at the pixel level, leveraging edge information extracted from visible images. By utilizing edges as contextual cues from the visible domain, the diffusion model achieves meticulous control over the delineation of objects within the generated images. To alleviate the impacts of those visible-specific edge information that should not appear in the thermal domain, a two-stage modality adversarial training (TMAT) strategy is proposed to filter them out from the generated images by differentiating the visible and thermal modality. Extensive experiments on LLVIP demonstrate ECDM's superiority over existing state-of-the-art approaches in terms of image generation quality. The pseudo thermal images generated by ECDM also help to boost the performance of various thermal object detectors by up to 7.1 mAP.

## CCS CONCEPTS

• **Computing methodologies** → **Neural networks**; *Object detection.*

## KEYWORDS

Diffusion model, Thermal image generation, Thermal object detection

**ACM Reference Format:**
Anonymous Authors. 2024. Data Generation Scheme for Thermal Modality with Edge-Guided Adversarial Conditional Diffusion Model. In *Proceedings of the 32nd ACM International Conference on Multimedia (MM'24), October 28-November 1, 2024, Melbourne, Australia.* ACM, New York, NY, USA, 9 pages. https://doi.org/10.1145/nnnnnnn.nnnnnnn

## 1 INTRODUCTION

In scenarios characterized by low-light or dark conditions, visible sensors often fail to yield substantial information, whereas

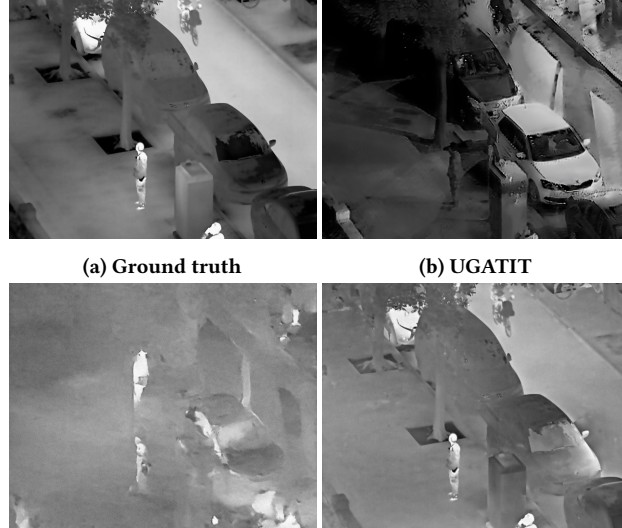

**Figure 1: A sample comparison of generated thermal images between different methods and ground truth. (a) A ground truth thermal image, (b) a generated thermal image by UGATIT, (c) a generated thermal image by DDIM and (d) a generated thermal image by ECDM (Ours).**

(a) Ground truth     (b) UGATIT

(c) DDIM     (d) ECDM (Ours)

thermal sensors capitalize on thermal radiation and temperature differentials. This sensitivity renders them particularly proficient in detecting temperature-related entities, notably livings and vehicles, within obscured settings. The superiority of thermal vision motivates numerous studies [10, 12, 17] dedicated to thermal vision applications, particularly thermal object detection, and yield noteworthy enhancements in this domain.

However, the efficacy of thermal vision applications remains notably curtailed by the paucity of available training samples. For example, the LLVIP dataset [16] only contains a mere 12,000 training thermal images, constituting only one-ninth of the visible samples contained within the COCO dataset [23]. The procurement of expansive training data and the meticulous labeling of precise annotations necessitate extensive human labor and substantial time investment. To address this challenge, extant studies mainly harness two methodologies for augmenting training datasets to facilitate deep model training: 3D synthesis and deep generative models. The 3D synthesis methods [3, 28] commence by generating a subset of 3D objects, followed by the application of a rudimentary thermal shader to render these objects, thereby engendering synthetic thermal images. Nevertheless, the images produced by the latest thermal sensor simulators still exhibit significant disparities when compared to those captured using real equipments. Recent studies in the domain of generative models, particularly within the realm

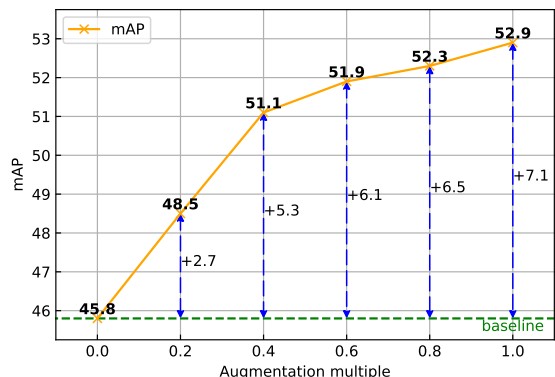

**Figure 2: The performance of RetinaNet trained with various amounts of generated pseudo training data. The x-axis indicates the augmentation multiple. For example, 0.2 indicates that the generated pseudo training data in the entire training sample is only 20% of the real data.**

of Generative Adversarial Networks (GANs) [15, 25], serves as an impetus for the generation of training data for thermal object detection [18, 26]. However, existing GAN-based methods necessitate the availability of paired visible and thermal images for training deep models. The constraint often proves challenging to meet in practical contexts.

This study delves into the applicability of the diffusion model to the task of generating thermally-aligned pixel-level images, obviating the need for supervisory guidance from paired visible-thermal counterparts. To this end, we propose an edge-guided adversarial condition diffusion model (ECDM) for thermal data generation. The basic idea of ECDM is learning the conditional probabilistic density of thermal images under the condition of the given visible image edge information. By incorporating the idea of adversarial learning to ease detrimental impacts stemming from extraneous and irrelevant edge details in the visible domain. As illustrated in Figure 1d, our ECDM can simultaneously reconstruct object shape and object thermal radiation characteristics while other methods are only good in one aspect. As shown in Figure 1b, GAN-based methods often tend to reconstruct object shapes but introduce irrelevant edges and abnormal details in thermal modality. As shown in Figure 1c, other Diffusion-based methods often tend to reconstruct object thermal radiation characteristics. In summary, the main contributions of ECDM are threefold:

(1) ECDM engenders pixel-level thermally-aligned images through the generation process, bypassing the necessity for annotated visible-thermal pairs. This innovation augments the available datasets for thermal object detection by effectively generating thermal images from their visible counterparts.

(2) We devise a conditional diffusion model to estimate the conditional probabilistic density of thermal images under the constraints of the given visible image edges. Furthermore, we develop an adversarial training strategy to filter out the

extraneous edge information from the visible domain that should not appear in the thermal domain.

(3) Extensive experiments on LLVIP demonstrate ECDM's superiority over existing state-of-the-art approaches in terms of image generation quality. The applicability of ECDM in generating training samples is also evaluated on the classical object detection task, wherein ECDM brings up to 7.1% mAP improvement for the detectors (as shown in Figure 2).

## 2 RELATED WORK

To solve the lacking of thermal images, some studies attempt to employ the domain adaptation techniques [17], by fine-tuning the pretrained visible object detectors into the thermal domain. The main promising ideas are multi-level feature alignments [29] and style consistency constraints [30]. Nonetheless, while domain adaptation methods may mitigate the issue of insufficient annotations, they continue to face challenges in the absence of thermal images.

Another mainstream of studies rely on generative-based methods to generate synthetic thermal images. In [2, 3, 28], there is a discussion of the use of virtual environments to create synthetic thermal images. These methods rely on intricate 3D models now focusing only on objects rather than whole scenes and employ infrared physics-based rendering. In [5, 18], generative models are discussed to create synthetic thermal images. But these generative models-based methods can not generate pixel-level alignments of thermal images from visible images.

Deep generative models (DGMs) are neural networks trained to approximate the probability distributions of data. After training successfully, we can generate new samples from the underlying distribution. Generative Adversarial Networks (GANs) [11], as a type of DGMs, have been extensively employed in the image-to-image translation tasks [15, 21, 24, 48]. Basic GANs consist of a generator and a discriminator under an adversarial training framework. The adversarial training process can be modeled as a min-max game. However, they have drawbacks such as poor convergence characteristics, especially on the thermal modality with rare textures.

Recently, diffusion models (DMs) [14] as a novel paradigm in the generative model, were shown impressive generative capabilities in high level of details [4]. Compared to GANs, this approach has a more stable training process and produces a greater range of diverse images. Recent advancements in DMs have demonstrated the ability to control the generation process, including details, through various conditions like image [20, 35, 45], class [6], and text [33]. DMs and their variants possess intriguing properties, such as stable training, generative diversity in images, and details control through conditions. These properties make them suitable for generating training data from visible images for the thermal object detection task.

Diffusion-GAN [40] attempt utilize the advantage of the flexible diffusion model to stability the training process of GANs. In contrast to [40] which injects adaptive noise via diffusion at various time steps to provide higher training stability over strong GAN baselines, our two-stage modality adversarial training (TMAT) strategy utilizes adversarial training to mitigate the distribution mismatch in generating images under diverse conditions.

Figure 3: Illustration of our Two-stage Modality Adversarial Training (TMAT) strategy. During the first stage, we only use $x^{tir}$ as input and train the ECDM to learn the distribution of $p_{model}(x^{tir}|\zeta^{tir})$. In the second stage, we use unpaired $x^{vis}$ and $x^{tir}$ as input and utilize GANs to reduce the gap between visible and thermal domains. This helps us learn the distribution of $p_{model}(x^{tir}|\zeta^{vis})$ for approximating $p_{tir}(x^{tir})$.

## 3 METHODOLOGY

### 3.1 Framework Overview

In this section, we first formalize the problem of *generating pseudo training data for thermal object detection*. As visible object detection datasets typically exhibit greater scale than their thermal counterparts, we leverage existing visible datasets to craft pseudo training samples for thermal object detection. Given a real visible object detection dataset $\mathcal{D}^{vis} = \{x_i^{vis}, y_i^{vis}\}_{i=1}^{N}$ contains $N$ visible images and a dataset $\mathcal{D}^{tir} = \{x_i^{tir}, y_i^{tir}\}_{i=1}^{M}$ contains $M$ real thermal infrared images, where each $x_i^{vis}, x_i^{tir}$ is an image sampled from a distribution $p_{vis}(x^{vis})$ or $p_{tir}(x^{tir})$, respectively. The corresponding annotations for each image are labeled as $y_i^{vis}$ and $y_i^{tir}$. In training generative models, the goal is to learn a model distribution $p_{model}(x^{tir}|x^{vis})$ that matches $p_{tir}(x^{tir})$.

However, different from the task of image generation or image-to-image translation, our generated pseudo images are prepared for thermal object detection. Consequently, generative thermal images necessitate pixel-level alignment with their corresponding visible images to accurately represent objects. To attain this precise alignment, we introduce the Edge Condition Diffusion Model (ECDM) rooted in conditional diffusion models. In our approach, edge images play a crucial role as the guiding condition during the sampling process for the creation of training samples.

Although edge information can bridge the thermal and visible domains, some discrepancies persist between the corresponding

edge images in these domains. To address this challenge, we introduce a two-stage modality adversarial training strategy instead of a direct end-to-end training approach for the ECDM. Initially, we utilize thermal edge images to train the ECDM, enabling it to translate thermal edge images into thermal images. Subsequently, we leverage the trained ECDM as a generator and devise a discriminator. Through adversarial training resembling a GAN approach, we work towards minimizing the disparity between synthetically generated thermal images under visible edge conditions and authentic thermal images.

### 3.2 Edge-guided Conditional Diffusion Model

Texture, shape, and color stand as the paramount visual cues in visual recognition [9]. However, due to the substantial differences in primary radiation, a considerable gap exists between thermal infrared and visible images. Specifically, thermal images have lack color information and exhibit lower texture than their visible counterparts. Shape information exhibits notable similarity between thermal and visible images within the same scene. We posit that this shape information serves as a bridge, mitigating the gap between the thermal and visible domains to some degree. The shape information in the images can be roughly extracted by high-frequency filtering

$$\zeta_i^m = \mathcal{H}(x_i^m), \tag{1}$$

where $m \in \{vis, tir\}$ is a modality indicator superscript, and $\mathcal{H}(\cdot)$ is an edge extracted operator contains a fast Fourier transform, a high

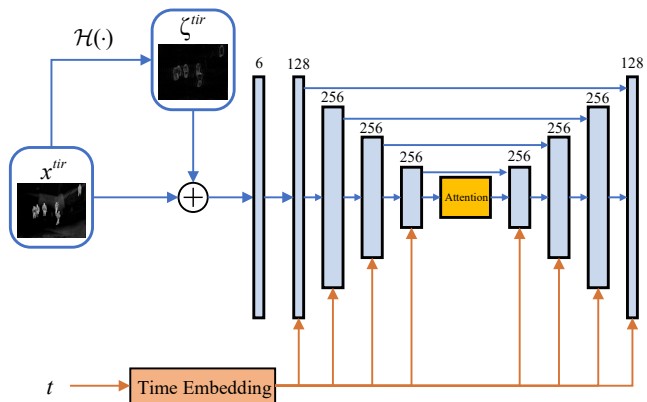

**Figure 4: The architecture of ECDM. The numbers over the light blue rectangle blocks denote the channels of feature maps. The yellow rectangle block denotes an attention layer.**

pass filtering, and an inverse fast Fourier transform. The extracted shape information is also termed as edge images. Different from image generation task in vanilla diffusion models [14, 37, 40] or conditioned on the class label, text, or natural images, the proposed ECDM introduces prior knowledge related to the fine-granularity content of objects, i.e., edge images.

The ECDM comprises both the diffusion process and the reverse process. It destroys the input thermal image $x_0^{tir} \in \mathcal{D}^{tir}$ to a standard Gaussian noise $x_T^{tir} \sim \mathcal{N}(0, \mathbf{I})$ by gradually adding small Gaussian noise in $T$ diffusion steps in forward process. The forward process of ECDM is presented as follows

$$q(x_{1:T}^{tir}|x_0^{tir}) = \prod_{t=1}^{T} q(x_t^{tir}|x_{t-1}^{tir}), \qquad (2)$$

$$q(x_t^{tir}|x_{t-1}^{tir}) \sim \mathcal{N}(x_t^{tir}; \sqrt{1 - \beta_t} x_{t-1}^{tir}, \beta_t \mathbf{I}), \qquad (3)$$

where $\beta_t$ is a small positive constant to control the variants of the added noise in diffusion step $t(0 \le t \le T)$. Following[14], the variance schedule is predefined linearly increasing from $\beta_1 = 10^{-4}$ to $\beta_T = 0.02$ and diffusion steps is seted $T = 1000$. The noised sample at diffusion step $t$ can be directly calculated by

$$x_t^{tir} = \sqrt{\bar{\alpha}_t} x_0^{tir} + \sqrt{1 - \bar{\alpha}_t} \kappa, \kappa \sim \mathcal{N}(0, \mathbf{I}), \qquad (4)$$

where $\alpha_t = 1 - \beta_t, \bar{\alpha}_t = \prod_{i=1}^{t} \alpha_i$.

The reverse process of ECDM is conditioned on the edge images to bridge the gap between the thermal domain and the visible domain while capturing the fine-granularity content of objects. Note that the different training stages use edge images in different domains. The reverse process is written as

$$p_\theta(x_{0:T-1}^{tir}|x_T^{tir}, \zeta^m) = \prod_{t=1}^{T} p_\theta(x_{t-1}^{tir}|x_t^{tir}, \zeta^m), \qquad (5)$$

$$p_\theta(x_{t-1}^{tir}|x_t^{tir}, \zeta^m) \sim \mathcal{N}(x_{t-1}^{tir}; \mu_\theta(x_t^{tir}, \zeta^m, t), \sigma_t^2 \mathbf{I}). \qquad (6)$$

The parameterizations of $\mu_\theta$ and $\sigma_\theta$ are defined by

$$\mu_\theta(x_t^{tir}, \zeta^m, t) = \frac{1}{\sqrt{\bar{\alpha}_t}} \left( x_t^{tir} - \frac{\beta_t}{\sqrt{1 - \bar{\alpha}_t}} \kappa_\theta(x_t^{tir}, \zeta^m, t) \right), \qquad (7)$$

$$\sigma_t^2 = \frac{1 - \bar{\alpha}_{t-1}}{1 - \bar{\alpha}_t} \beta_t, \qquad (8)$$

where $\kappa_\theta$ is a neural network parameterized by $\theta$, implemented by a modified UNet [14, 34]. In our work, $\kappa_\theta$ takes the noised image $x_t^{tir}$, the conditional edge image $\zeta^m$, and the diffusion step $t$ as input. The the diffusion step $t$ are fed into a Transformer sinusoidal position embedding layer [39] with a given embedding dimension, followed by a Linear + sigmoid layer. Then, for each downsample and upsample block, an additional Linear + sigmoid layer is employed to align the channel dimensions of the feature maps. Figure 4 shows the architecture of ECDM.

### 3.3 Two-stage Modality Adversarial Training

Our goal is to learn the distribution of $p_{model}(x^{tir}|\zeta^{vis})$. However, attempting to manually train the ECDM in a straightforward end-to-end manner, such as $\zeta^m = \zeta^{vis}$ in (7), while theoretically feasible, becomes challenging in practice due to the substantial divergence between the thermal and visible domains.

To address this challenge, we propose a two-stage modality adversarial training strategy. As illustrated in Figure 3, in the first stage, we set $\zeta^m = \zeta^{tir}$ and train the model to learn the distribution of $p_{model}(x^{tir}|\zeta^{tir})$. The condition $\zeta^{tir}$ and generative images $x^{tir}$ all remain within the thermal domain, simplifying the distribution learning process. In the second stage, we incorporate the principles of GANs, employing the ECDM previously trained in the first stage as the generator and employing a thermal modality authenticity indicator as the discriminator. At this stage, we begin by generating images from the generator under a distinct condition $\zeta^m = \zeta^{vis}$, which introduces modality bias due to the incongruity between $p(\zeta^{tir})$ and $p(\zeta^{vis})$. We then mitigate this modality bias through adversarial training. The complete two-stage modality adversarial training procedure is detailed in Algorithm1.

Specifically, in the first stage, we focus on training the ECDM by optimizing the standard variational bound on negative log-likelihood. Following the reparameterization trick in [14], the training objective of ECDM in the first stage training process is

$$\mathcal{L}_{diff} = \mathbb{E}_{x_0^{tir}, \kappa, t} \parallel \kappa - \kappa_\theta(x_t^{tir}, \zeta^{tir}, t) \parallel_2^2 . \qquad (9)$$

In the second stage, we leverage the ECDM as a generator $G$ and introduce a discriminator $D$ by PatchGAN [15], where $G$ aims to generate synthetic thermal images $\hat{x}^{vis} \sim p_{tir}(x^{tir})$ the condition of visible edge images, $D$ aims to distinguish the real and synthetic thermal images. To ensure that the generated thermal images are indistinguishable from authentic ones, we employ an adversarial loss [11]

$$\mathcal{L}_{adv} = \lambda_{real} \mathbb{E}_{x^{tir}} [\log(D(x^{tir}))] + \mathbb{E}_{\kappa, \zeta^{vis}} [\log(1 - D(G(\kappa, \zeta^{vis})))], \qquad (10)$$

where $\lambda_{real}$ is a hyper-parameter to balance the adversarial loss components. We set $\lambda_{real} = 10$ in our experiments. To further narrow the gap between the generated thermal images and actual ones, we incorporate a style-consistency loss defined as

$$\mathcal{L}_{style} = \|\hat{x}^{tir} - x^{tir}\|_2^2, \qquad (11)$$

and design a modality-consistency loss

$$\mathcal{L}_{mod} = \|1 - D(\hat{x}^{vis})\|_2^2, \qquad (12)$$

**Algorithm 1** Two-stage Modality Adversarial Training

---

**Require:** the thermal image $x^{tir}$, the visible images $x^{vis}$, the first stage training epochs $S_{diff}$, the second stage training epochs $S_{adv}$, the steps of generating $S_G$, the steps of discriminating $S_D$.

1: $x_T \sim \mathcal{N}(0, \mathbf{I})$
2: $\zeta^{tir} = \mathcal{H}(x^{tir}), \zeta^{vis} = \mathcal{H}(x^{vis})$
3: **for** $i = 0$ to $S_{diff}$ **do**
4:      Update $\kappa_\theta$ by descending its gradient:
5:      $\nabla_{\kappa_\theta} \parallel \kappa - \kappa_\theta(x_t^{tir}, \zeta^{tir}, t) \parallel_2^2$
6: **end for**
7: $\hat{x}^{tir} \leftarrow G(\kappa, \zeta^{tir}), \hat{x}^{vis} \leftarrow G(\kappa, \zeta^{vis})$
8: **for** $j = 0$ to $S_{adv}$ **do**
9:      **for** $k = 0$ to $S_G$ **do**
10:         Update $\kappa_\theta$ by descending its gradient:
11:         $\nabla_{\kappa_\theta} \parallel \kappa - \kappa_\theta(x_t^{tir}, \zeta^{tir}, t) \parallel_2^2$
12:         $\nabla_{\kappa_\theta} \|\hat{x}^{tir} - x^{tir}\|_2^2$
13:         $\nabla_{\kappa_\theta} \|1 - D(\hat{x}^{vis})\|_2^2$
14:         $\nabla_{\kappa_\theta} \|\mathcal{H}(x^{vis}) - \mathcal{H}(\hat{x}^{vis})\|_2^2$
15:      **end for**
16:      **for** $l = 0$ to $S_D$ **do**
17:         Update $D$ by descending its gradient:
18:         $\nabla_D [\lambda_{real} \| \log(D(x^{tir})) \|_2^2$
19:             $+ \| \log(1 - D(\hat{x}^{vis})) \|_2^2]$
20:      **end for**
21: **end for**
22: **return** ECDM ($\kappa_\theta$).

---

where $\hat{x}^{tir} := G(\kappa, \zeta^{tir})$ and $\hat{x}^{vis} := G(\kappa, \zeta^{vis})$ represent synthetic thermal images under conditions of thermal edge images or visible edge images, respectively. Additionally, we utilize an edge loss to preserve accurate boundaries and highly detailed shapes, which is defined as

$$\mathcal{L}_{edge} = \|\mathcal{H}(x^{vis}) - \mathcal{H}(\hat{x}^{vis})\|_2^2. \tag{13}$$

Finally, we express the objective functions to optimize $G$ and $D$, respectively, as follows

$$\mathcal{L}_G = \lambda_{diff}\mathcal{L}_{diff} + \lambda_{style}\mathcal{L}_{style} + \lambda_{mod}\mathcal{L}_{mod} + \lambda_{edge}\mathcal{L}_{edge}, \tag{14}$$

$$\mathcal{L}_D = \mathcal{L}_{adv}, \tag{15}$$

where $\lambda_{diff}, \lambda_{style}, \lambda_{mod}, \lambda_{edge}$ are hyper-parameters to control the weight of different loss. We use $\lambda_{diff} = 0.1, \lambda_{style} = 100, \lambda_{mod} = 1, \lambda_{edge} = 1000$ in all our experiments. Moreover, we use dpm-solver++ [27] to improve the sampling speed of ECDM. The sampling parameters of DPM-solver++ are listed in the supplementary material.

## 4 EXPERIMENTS

### 4.1 Experimental Settings

*4.1.1 Datasets.* Our experiments are mainly on the Low-Light Visible-Infrared Paired Dataset (LLVIP) [16], FLIR thermal dataset (FLIR) [8], and Person Re-identification in the Wild Dataset (PRW) [47]. LLVIP consists of 15,488 pairs of visible-thermal images, captured under low-light conditions using a binocular surveillance camera.

These paired images are precisely aligned both spatially and temporally. For brevity, we refer to this dataset as $\mathcal{D}_{llvip}$. FLIR has two versions: v1.3 (2019) and v2.0 (2021). We utilize v2.0 for its expanded training dataset. Our analysis focuses on five categories: person, bike, car, light, and sign. PRW comprises 11,816 frames captured during the summer months using a visible camera. These frames were extracted from the Market-1501 dataset [46]. This dataset is annotated for both person re-identification and pedestrian detection tasks. We refer to this dataset $\mathcal{D}_{prw}$.

*4.1.2 Metrics.* We use Fréchet Inception Distance (FID) [13], Learned Perceptual Image Patch Similarity (LPIPS) [44], peak-noise-to-signal ratio (PSNR), the structural similarity index measure (SSIM), and Kernel Inception Distance (KID) [1] to measure the quality of generated thermal images. FID is a widely adopted non-reference perceptual metric that assesses the similarity between two sets of images.

We denote the standard implementation in [36] as FID. Besides, the FID implemented in [31] is FID-C, and the FID implemented using CLIP instead of InceptionV3 is FID-C$_{clip}$. Additionally, we employ KID, a metric similar to FID but with a polynomial kernel for an unbiased estimator.

We use standard mean Average Percision [23] (mAP) under different Intersection over Union (IoU) thresholds as the metrics to measure the gain of generated pseudo training data to the performance of thermal detectors.

### 4.2 Implementation Details

We train the ECDM on four NVIDIA 3090 24GB GPUs, utilizing a batch size of 4 and resizing the input images to a resolution of $512 \times 640$. The generator $G$ and discriminator $D$ are optimized using Adam with $\beta_1 = 0.9$ and $\beta_2 = 0.999$. The learning rate is set to 0.00002. Our training procedure involves setting $S_{diff} = 70$, $S_{adv} = 20$, $S_G = 2$, and $S_D = 1$. Further details can be found in the supplementary material.

Our training process in Sec. 4.7 utilizes two NVIDIA 3090 24GB GPUs with a batch size of 32 and employs SGD as the optimizer. We set the base learning rate to 0.0002, momentum to 0.99, and weight decay to 0.0001. To ensure a fair comparison, we train them for 40k iterations and use a multistep learning rate scheduler with steps at 12000, 18000, and 32000 iterations.

### 4.3 Quantitative and Qualitative Comparison to Visible-to-thermal Translation Task

We demonstrate that ECDM can deliver competitive results in visible-to-thermal translation tasks. We compare our method with several state-of-the-art methods: pix2pixGAN [15], CycleGAN [48], UGATIT [19], LPTN [21], VSAIT [38], DDIM [37] and BBDM [20]. We also benchmark some energy-based/flow-based models in our experiments, but their performance was inferior to the GAN-based methods presented in Table 1. We report the performance under $512 \times 640$ resolution which specifically addresses thermal object detections. For DDIM, we modified the original architecture as shown in Figure 4. However, in our case, we used visible images from the LLVIP dataset as the conditioning input for the visible-to-thermal image translation task. We evaluate BBDM's performance by up-sampling generated images from 256×256, due to its difficulty in converging at 512×640.

As can be seen in Table 1, ECDM achieves superior performance to SOTA thermal image generators on the LLVIP dataset. It suggests that by introducing visible edge conditions to guide diffusion modeling, our proposed ECDM effectively generates high-quality pixel-level aligned pseudo thermal images with visible edge images.

**Table 1: Quantitative comparison on the LLVIP dataset. $^{\S}$ indicates upsampling from the 256×256 resolution. Best results highlighted in bold, second best in underline.**

| Method | FID↓ | LPIPS↓ | SSIM↑ | PSNR↑ |
|---|---|---|---|---|
| pix2pixGAN (CVPR 2017) | 317.38 | 0.474 | 0.211 | 11.251 |
| CycleGAN (ICCV 2017) | 183.80 | 0.354 | 0.283 | 12.196 |
| UGATIT (ICLR 2020) | 178.71 | 0.359 | 0.285 | 12.970 |
| LPTN (CVPR 2021) | 209.84 | 0.396 | 0.245 | 11.658 |
| VSAIT (ECCV 2022) | 211.30 | 0.360 | 0.277 | 13.050 |
| DDIM (ICLR 2021) | 325.87 | 0.454 | 0.393 | 11.741 |
| BBDM$^{\S}$ (CVPR 2023) | 265.06 | 0.436 | 0.311 | 11.728 |
| ECDM (Ours) | **139.91** | **0.141** | **0.507** | **13.130** |

We also show qualitative comparison with other methods in Figure 5. Our qualitative comparison is based on two key principles: **object shape reconstruction** and **object thermal radiation characteristic reconstruction**. The latter principle is based on the thermal imaging mechanism. Thermal imaging relies on emitting radiation from objects. The higher the temperature, the brighter the object in the thermal image.

In Figure 5, although some methods can reconstruct full objects like cars, they still fail to capture the thermal radiation characteristic of objects in generated images. For example, the tires of moving vehicles and the exposed skin of pedestrians should appear brighter in thermal images. GAN-based methods, such as CycleGAN and UGATIT, excel in object shape reconstruction but fall short in object thermal radiation characteristic reconstruction. These GAN-based methods also tend to reconstruct the object texture rather than the object thermal radiation characteristics, such as stationary vehicles and sidewalks. Conversely, diffusion-based methods, such as DDIM and BBIM, demonstrate proficiency in thermal radiation characteristic reconstruction but struggle with shape reconstruction. Our proposed method stands out by achieving both shape reconstruction and thermal radiation characteristic reconstruction simultaneously. More visual results can be found in supplementary material.

## 4.4 Model Complexity Comparison

We compare the complexity of ECDM with other methods in terms of the number of parameters (#params) and FLOPs. All results computed using the calflops package [41].

As can be seen in Table 2, our method is comparable to GAN-based method except LPTN in #params. Compared with DDIM, our method introduces an additional discriminator, resulting in an increase of 2.765M #params. Diffusion-based methods generally require higher FLOPs because of multisteps reversal process compared with GAN-based methods. However, our method benefits

**Table 2: Comparison of model complexity and parameters. #params denotes the number of parameters. G and T after the values represent the unit of FLOPs. Different colors are used for better distinguish.**

| Type | Method | #params (M) ↓ | FLOPs ↓ |
|---|---|---|---|
| GAN-based | pix2pixGAN | 60.290 | 44.598G |
| | CycleGAN | 28.286 | 496.415G |
| | UGATIT | 32.946 | 134.577G |
| | LPTN | 0.871 | 13.629G |
| | VSAIT | 65.492 | 642.878G |
| Diffusion-based | DDIM | 34.431 | 733.246T |
| | BBDM | 273.095 | 806.380T |
| | ECDM (Ours) | 37.196 | 120.986T |

from the application of DPM-Solver++, which results in the lowest FLOPs among diffusion-based methods.

We also report that generating $512 \times 640$ of one thermal image needs about 14.7s on a NVIDIA 3090 GPU. However, we can apply some acceleration schemes of diffusion to tackle the efficiency problem. Besides, the data generation is a one-time operation, and any other downstream tasks can use them.

## 4.5 Transferability of ECDM

In the experimental setup described above, the visible edge images and target generated thermal images are from the same dataset (all in the LLVIP dataset), resulting in a small gap between them. In practice, generating pseudo thermal images that can yield gains for downstream tasks such as thermal object detection poses a challenging problem: will the generated training data be useful when the conditions are far from the target domain? This raises the issue of model transferability.

**Table 3: Ablation study for transferability of ECDM.**

| Condition | | FID↓ | FID-C↓ | FID-C$_{clip}$ ↓ | KID↓ |
|---|---|---|---|---|---|
| Training | Sampling | | | | |
| $\mathcal{D}_{llvip}^{tir}$ | $\mathcal{D}_{prw}$ | 306.66 | 305.00 | 66.59 | 0.2881 |
| $\zeta_{llvip}^{tir}$ | $\zeta_{prw}$ | 249.49 | 245.84 | 36.84 | 0.2265 |
| $\mathcal{D}_{llvip}^{vis}$ | $\mathcal{D}_{prw}$ | 267.59 | 264.56 | 41.43 | 0.2744 |
| $\zeta_{llvip}^{vis}$ | $\zeta_{prw}$ | 278.38 | 280.99 | 44.75 | 0.3198 |
| $\mathcal{D}_{prw}$ | $\mathcal{D}_{prw}$ | 294.32 | 285.56 | 38.85 | 0.3255 |
| $\zeta_{prw}$ | $\zeta_{prw}$ | 266.97 | 267.29 | 44.35 | 0.2997 |

To evaluate the transferability of ECDM, we train our model under various conditions, including thermal image conditions (denoted as $\mathcal{D}_{llvip}^{tir}$), thermal edge conditions (denoted as $\zeta_{llvip}^{tir}$), visible images in the LLVIP dataset (denoted as $\mathcal{D}_{llvip}^{vis}$), visible edge images in the LLVIP dataset (denoted as $\zeta_{llvip}^{vis}$), and visible images in the PRW dataset (denoted as $\mathcal{D}_{prw}$), visible edge images in the PRW dataset (denoted as $\zeta_{prw}$). We directly sample thermal images under $\mathcal{D}_{prw}$ or $\zeta_{prw}$ conditions from the aforementioned cases. As

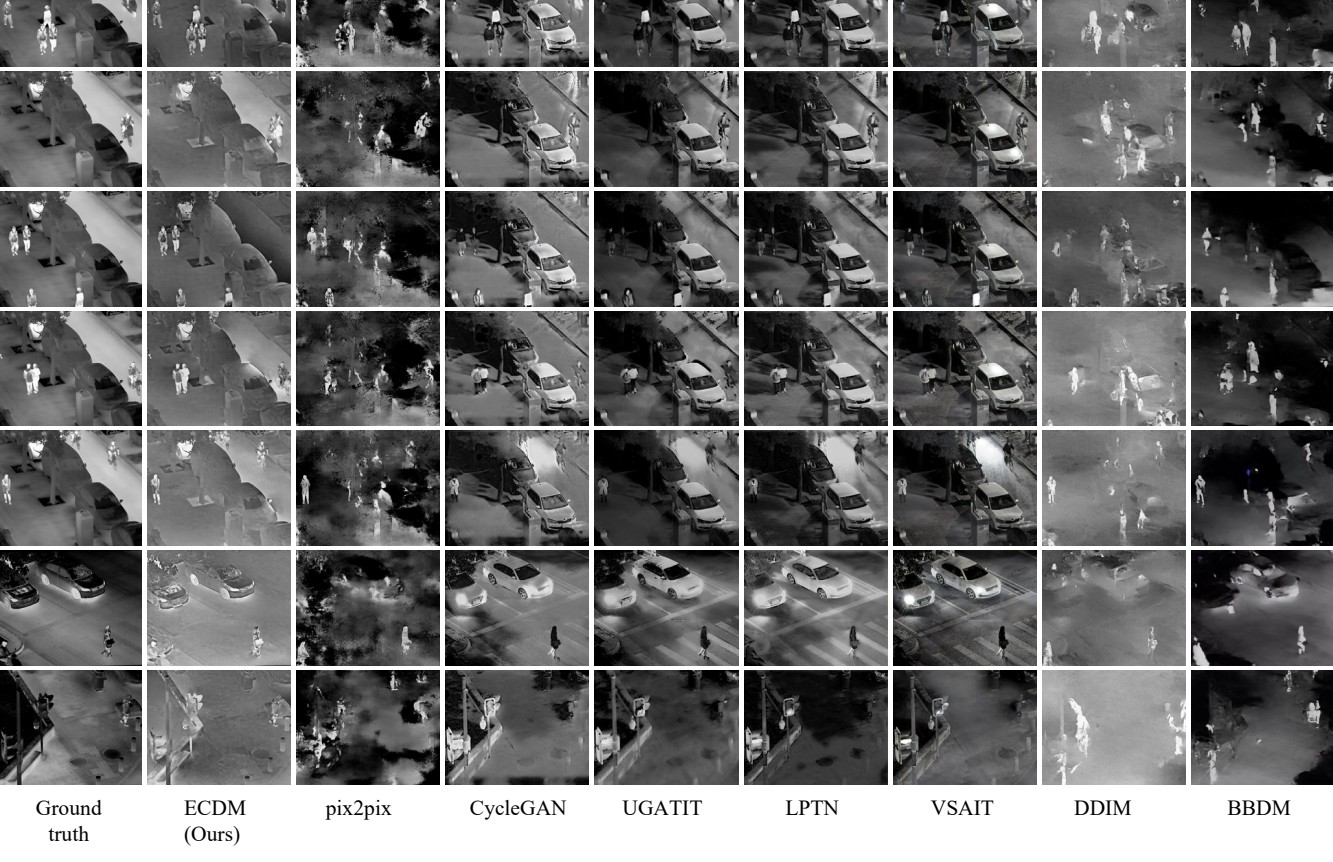

| Ground truth | ECDM (Ours) | pix2pix | CycleGAN | UGATIT | LPTN | VSAIT | DDIM | BBDM |

**Figure 5: Qualitative comparison of our proposed method with other state-of-the-art methods on the LLVIP test dataset. To ensure fairness and randomness, we use Python's random module with a fixed seed (1234) to select four images from the dataset. The selected images are '190145.jpg', '190345.jpg', '190373.jpg', '190405.jpg', '190480.jpg', '220224.jpg', '260261.jpg'. More visual results can be found in supplementary material.**

shown in Table 3, substantial degradations in various metrics are evident due to domain and dataset disparities. Nonetheless, well-trained ECDM attains a minimal FID-C score under cross-domain and cross-dataset sampling conditions.

## 4.6 Effects of TMAT with Different Edge Information

To validate the effectiveness of our TMAT strategy in the face of transferability, we train the ECDM with TMAT under diverse edge inputs. As shown in Table 4, the trends of various metrics are consistent, demonstrating that TMAT diminishes the gap between disparate domains and datasets.

## 4.7 A Showcase of ECDM on Thermal Object Detection

To investigate the impact of the number of pseudo thermal images on thermal object detection performance, we select RetinaNet [22] as our baseline and train it under the same settings, except for the training data. The training data consists of two parts: real thermal

**Table 4: Ablation study for TMAT.**

| Condition | TMAT | FID↓ | FID-C↓ | FID-C$_{clip}$ ↓ | KID↓ |
|---|---|---|---|---|---|
| Thermal edge images | ✗ | 249.49 | 245.84 | 36.84 | 0.2265 |
|  | ✔ | 248.36 | 241.96 | 38.98 | 0.2292 |
| Visible edge images (night time) | ✗ | 278.38 | 280.99 | 44.75 | 0.3198 |
|  | ✔ | 264.70 | 259.63 | 37.02 | 0.2499 |
| Visible edge images (day time) | ✗ | 266.97 | 267.29 | 44.35 | 0.2997 |
|  | ✔ | 258.59 | 251.82 | 43.18 | 0.2485 |

images from the training set of LLVIP and pseudo thermal images generated by our ECDM using PRW edge images. The number of real thermal images is fixed in the training data and the number of pseudo thermal images generated by our ECDM is controled by the augmentation multiple ratio. For instance, an augmentation multiple ratio of 0.2 signifies that the generated pseudo training data constitutes only 20% of the real data in the entire training set. We experiment with diverse augmentation multiple ratios, namely 0, 0.2, 0.4, 0.6, 0.8, and 1.0, and observe the impact on mAP. As

depicted in Figure 2, the mAP improves gradually from 0 to 7.1, with the most significant enhancement occurring at a ratio of 1.0.

To further effective the generalization of pseudo training data generated by our ECDM, we train various object detectors, including Faster RCNN [32], RetinaNet [22], CenterNet [7], VFNet [43], and DINO [42]. For a fair comparison, we maintain an augmentation multiple ratio of 1.0 throughout this experiment.

Our generated pseudo training data yield mAP improvements ranging from 1.1 (CenterNet) to 7.1 (RetinaNet) across different detectors on the LLVIP dataset. Notably, most detectors, excluding VFNet, exhibit mAP improvements between 0.7 and 1.8, demonstrating the effectiveness of our pseudo training data on the FLIR dataset.

**Table 5: Using pseudo data training different detectors on LLVIP dataset and FLIR dataset. The red color means performance improvement while the green color represents performance decrease.**

| Method | Batch size | Backbone | Dataset | Pseudo data | mAP | mAP@50 | mAP@75 |
|---|---|---|---|---|---|---|---|
| Faster RCNN | 32 | Resnet-50 | LLVIP | ✗ | 49.0 | 89.2 | 48.4 |
| | | | | ✔ | 50.3 (+1.3) | 89.2 | 51.6 |
| | | | FLIR | ✗ | 23.5 | 42.7 | 22.6 |
| | | | | ✔ | 24.4 (+0.9) | 44.0 | 23.1 |
| RetinaNet | 32 | Resnet-50 | LLVIP | ✗ | 45.8 | 90.3 | 40.7 |
| | | | | ✔ | 52.5 (+7.1) | 92.8 | 53.6 |
| | | | FLIR | ✗ | 14.5 | 29.4 | 12.4 |
| | | | | ✔ | 15.2 (+0.7) | 30.8 | 13.2 |
| CenterNet | 32 | Resnet-50 | LLVIP | ✗ | 53.4 | 91.3 | 56.3 |
| | | | | ✔ | 54.5 (+1.1) | 93.1 | 57.7 |
| | | | FLIR | ✗ | 25.5 | 48.9 | 22.8 |
| | | | | ✔ | 27.3 (+1.8) | 52.1 | 24.5 |
| VFNet | 32 | Resnet-50 | LLVIP | ✗ | 52.2 | 91.3 | 54.1 |
| | | | | ✔ | 54.7 (+2.5) | 92.9 | 58.6 |
| | | | FLIR | ✗ | 15.1 | 32.0 | 12.2 |
| | | | | ✔ | 13.0 (-2.1) | 27.8 | 10.7 |
| DINO | 2 | Swin-L | LLVIP | ✗ | 40.2 | 72.9 | 39.8 |
| | | | | ✔ | 44.2 (+4.0) | 74.9 | 46.9 |
| | | | FLIR | ✗ | 7.6 | 16.0 | 6.5 |
| | | | | ✔ | 9.2 (+1.6) | 20.8 | 6.8 |

The Varifocal Loss in VFNet focuses the training on those high-quality positive examples that are more important for achieving a higher AP than those low-quality ones [43]. Considering the dataset characteristics, FLIR comprises objects of different scales, with a predominant presence of small-scale objects. LLVIP primarily consists of medium-scale objects, while PRW contains a mix of medium- and large-scale objects. The unique design of the Varifocal Loss in VFNet places more emphasis on the generated pseudo objects compared to other detectors since large-scale objects are more easily identified as positive examples than small-scale objects. This design also explains the performance decrease of VFNet on the FLIR dataset and its performance improvement on the LLVIP dataset.

We also evaluate the performance of pseudo-training data generated by ECDM or other image-to-image translation techniques on the RetinaNet. As shown in Table 6, when compared to the absence of any additional training data, the utilization of generated pseudo-training data proves effective in enhancing the performance of RetinaNet. Notably, our ECDM achieves the highest improvement (+7.1 mAP) compared with other methods.

**Table 6: Comparing the impact of pseudo training data generated by different methods on the performance of RetinaNet. 'None' indicates that no generated training data is used, while 'PRW' denotes the utilization of the PRW dataset as additional training data. Best results highlighted in bold, second best in underline.**

| Generating Method | mAP | mAP@50 | mAP@75 |
|---|---|---|---|
| None (baseline) | 45.8 | 90.3 | 40.7 |
| PRW | 48.0(+2.2) | 91.4 | 44.8 |
| CycleGAN | 51.9(+6.1) | 92.6 | 52.9 |
| UGATIT | 51.1(+5.3) | 89.7 | 53.8 |
| LPTN | 52.4(+6.6) | 92.0 | 54.4 |
| VSAIT | 52.6(+6.8) | 93.0 | 54.6 |
| DDIM | 50.6(+4.8) | 92.4 | 49.9 |
| ECDM (Ours) | **52.9(+7.1)** | 92.7 | 55.3 |

## 5 LIMITATIONS AND FUTURE WORKS

Our ECDM currently requires high-quality edge images, which limits its applicability in certain scenarios. Furthermore, we also noticed that the generated images often have global color levels error, especially on the ground, traffic lights, and woods, which may account for the large FID score in Table 1. Additionally, as illustrated in Fig 5, our method and DDIM tend to generate brighter thermal images compared to the ground truth or the CycleGAN result. Notably, BBDM, also a diffusion-based method, does not exhibit this trend. Considering the comparison of model parameters in Table 2. We attribute this issue to the U-Net's limitations in parameter count and learning capacity.

Considering the relationship between the thermal object detection and other downstream tasks which also suffer the lacking of training samples like thermal object tracking, thermal semantic segmentation, and thermal image super resolution, our method is expected to bring performance improvements to them. However, tracking tasks typically require image sequences as input, necessitating the design of additional modules to ensure frame-inter consistency for generated images. It also exhibits potential applicability to image-to-image translation where edges can act as connectors between the source and target domain, particularly when the texture information in the target domain is relatively sparse.

## 6 CONCLUSION

In this paper, we introduce a novel data generation scheme for the thermal modality, called ECDM, which leverages a diffusion model to generate pixel-level aligned thermal images. Our approach utilizes edge images extracted from visible images as a condition to guide the diffusion model in learning the fine control of object boundaries in the generated image. To address the domain gap between thermal and visible images, we propose TMAT, a method that trains our ECDM to generate thermal images from visible edge images. Our extensive experiments demonstrate the promising performance of ECDM, and we conduct an exhaustive ablation study to analyze its effectiveness.

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
