# OpenReview forum: "Data Generation Scheme for Thermal Modality with Edge-Guided Adversarial Conditional Diffusion Model"
_acmmm.org/ACMMM/2024/Conference — MM2024 Poster_

### Official Review · Reviewer_AWBx · 2024-05-20

**Rating:** 4
**Confidence:** 3

**Summary:**

This paper investigates the generation of thermal modality data. The author proposes to utilize the edge information to bridge the domain gap between real vision images and thermal modality data, and designs an adversarial training strategy to filter out irrelevant edge information. The overall logic of the paper is smooth, and the presentation is clear. The model design is scientifically reasonable. The experiments are enough and effectively validate the effectiveness of the method design. The generated results show significant improvements compared to previous methods.

**Strengths:**

1.	The paper's logic is smooth, and the expression is clear. The motivation is reasonable, providing a comprehensive analysis of the existing problems in the current field.
2.	The design demonstrates a certain level of innovation, effectively addressing the issues raised in the article. Additionally, it shows significant improvement compared to other works in this field.
3.	The experiments are adequate and complete, strongly demonstrating the effectiveness and reasonableness of the model design, effectively supporting the article's conclusions.

**Limitations:**

1.	The authors propose a two-stage method, with the first stage conducted entirely on thermal modality data and the second stage incorporating real vision data through adversarial training. A potential issue is whether the training in the second stage might degrade the model obtained from the first stage. This point is not addressed in the paper, and it would be helpful if the authors could provide clarification on this matter.
2.	The paper primarily describes the training process but does not detail the sampling process. Therefore, I assume the sampling process follows the sampling way of the basic diffusion model, from random noise to images. Since there is no condition in the diffusion process to control generation, it raises the question of whether the content generated by the proposed method might include some anti-fact elements. It would be beneficial if the authors could provide more generated samples to illustrate this.

**Suitability:**

3

---

### Official Review · Reviewer_ZDWB · 2024-05-21

**Rating:** 3
**Confidence:** 3

**Summary:**

This article proposes an edge guided conditional diffusion model (ECDM) for generating pseudo thermal images aligned with visible light images at the pixel level to enhance the performance of thermal imaging algorithms in areas such as object detection. A two-stage Modal Adversarial Training (TMAT) strategy was introduced, which trains the model on hot edge images first, and then uses adversarial training to reduce the distribution difference between visible light and thermal imaging domains, thereby improving the quality of generated images. The superiority of ECDM in image generation quality has been verified through extensive experiments.

**Strengths:**

1. The TMAT strategy gradually reduces the distribution difference between the generated domain and the target domain through phased training, which is a novel and effective idea.
2. The experimental design of the article is relatively sufficient and has achieved significant improvements in multiple indicators, verifying the superiority of ECDM in image generation quality.
3. The article has the potential for widespread application, as this method is not only suitable for object detection, but also has the potential to be applied to other scenarios that require thermal imaging data, such as thermal imaging super-resolution, thermal target tracking, etc.

**Limitations:**

1. The article lacks comparison with professional SOTA papers, making it difficult to demonstrate its reliability. Pix2pix and cyclegan are very early articles, and recent gan-based articles such as LPTN and VSAIT are not specifically targeted for image translation in the fields of infrared thermal and visible light image conversion. This greatly undermines the superiority of the article and makes it difficult for practitioners in the infrared field to fully believe it.
2. The Figure 3 of the article lacks sufficient textual explanations or legends, and the lines and various modules lack explanations. Readers may need to put in more effort to understand each component in the figure and their relationships.
3. There may be an error in Figure 3 of the article. Formula (11) indicates that in order to further narrow the gap between the generated thermal image and the actual thermal image, the style consistency loss in Figure 3 is illustrated as the binomial between the actual thermal image after high-frequency sampling and the generated thermal image.

**Suitability:**

2

---

### Official Review · Reviewer_CReY · 2024-05-24

**Rating:** 3
**Confidence:** 4

**Summary:**

This paper introduces a novel infrared data generation method based on diffusion models, and employs a Generative Adversarial Network adversarial training strategy to mitigate the issue of excessive domain disparity during the generation process. The generated infrared images from this method exhibit a superior recovery of thermal map information compared to existing generation techniques, and additionally, they enhance the performance of downstream perception tasks.

**Strengths:**

This paper introduces a novel infrared data generation method based on diffusion models, and employs a Generative Adversarial Network adversarial training strategy to mitigate the issue of excessive domain disparity during the generation process. The generated infrared images from this method exhibit a superior recovery of thermal map information compared to existing generation techniques, and additionally, they enhance the performance of downstream perception tasks.

**Limitations:**

1. Firstly, I believe the motivation of the paper needs reconsideration. If the objective is to enhance perception tasks in adverse environments (low-light, rain, haze) where infrared imaging is necessitated, then the available visible image texture information would presumably be minimal. On the other hand, if the aim is to enhance the performance of downstream perception tasks under infrared vision algorithms, why not consider fusion the details of visible image texture information during generation? A generation approach akin to multimodal fusion could potentially bolster the performance of downstream tasks more effectively.
2. Why only utilize datasets of infrared and visible light under low-light conditions? There are now more diverse environments captured in datasets like M3FD and FMB, which could provide a broader basis for evaluation. And this kind of data set is more suitable for motivation.
3. Regarding the comparison with diffusion-based methods such as DDIM and BBDM, are these strategies trained using the proposed TMAT approach? Utilizing DDIM solely for accelerating sampling should not lead to significant performance degradation. Additionally, I am curious about the performance comparisons between DPM solver and other sampling accelerators.
4. This paper should give some visible results of detection to provide a clearer demonstration of the effectiveness of the proposed methods.

**Suitability:**

3

---

### Official Review · Reviewer_w9Da · 2024-05-25

**Rating:** 4
**Confidence:** 1

**Summary:**

This paper generates thermal images using edge-guided conditional diffusion model and two-stage modality adversarial training, in spirit combining diffusion models and GANs.

**Strengths:**

1. The results are promising. Experimental results show that the proposed method is effective and brings evident gains.

2. Table 5 shows the effectiveness of the generated data for downstream tasks.

3. The two-stage modality adversarial training is interesting.

**Limitations:**

1. In the experiments, the proposed method should be compared to more state-of-the-art diffusion models. Only comparing to DDIM and BBDM can be a bit limited.

2. In Table 6, the gain margin of the proposed method is not very evident.

3. Figure 3 can be improved in terms of clarity of the illustration.

4. The edge-guided conditional diffusion model is not very novel.

**Suitability:**

2

---

### Meta-Review · Area_Chair_yfii · 2024-07-07

**Recommendation:** Accept (Poster)
**Confidence:** 4

**Metareview:**

The paper originally received mixed ratings of BA, BR, BR and BA respectively. The reviewers were generally appreciative of the innovative method and the novel TMAT strategy, promising results and effectiveness of the generated data for downstream tasks, potential of wider application beyond object detection, adequate experimental details etc. However, they also raised concerns on the lack of comparisons with more SOTA diffusion models, clarity of motivation and aim of the proposed work, missing performance comparisons between DPM solver and other sampling accelerators, missing details of sampling process employed and possibility of anti-facts in the generated images, missing important details in Figure 3 etc. The authors have provided a detailed rebuttal with an effort to clarify most of the concerns raised. The reviewers w9Da, CReY and AWBx comment that the rebuttal has been able to partially/fully clarify their concerns, whereas reviewer ZDWB still has concern regarding comparison with more SOTA methods. Finally, three out of the four reviewers have retained their original rating, whereas reviewer CReY has upgraded the rating from BR to BA.

Upon careful consideration of the original reviews, authors' rebuttal and the post-rebuttal ratings (BA, BA, BR, BA), the AC feels that the merit in the contributions of the paper as well as its potential wide utility outweigh the remaining limitations and hence recommends acceptance. The authors are strongly encouraged to take all the reviewers' comments into account while preparing the camera-ready submission.